# Unsupervised Multi-view Pedestrian Detection

Mengyin Liu
blean@live.cn
School of Computer and Communication Engineering,
University of Science and Technology Beijing
Beijing, China

Chao Zhu*
chaozhu@ustb.edu.cn
School of Computer and Communication Engineering,
University of Science and Technology Beijing
Beijing, China

Shiqi Ren
shiqiren@xs.ustb.edu.cn
School of Computer and Communication Engineering,
University of Science and Technology Beijing
Beijing, China

Xu-Cheng Yin
xuchengyin@ustb.edu.cn
School of Computer and Communication Engineering,
University of Science and Technology Beijing
Beijing, China

## Abstract

With the prosperity of the intelligent surveillance, multiple cameras have been applied to localize pedestrians more accurately. However, previous methods rely on laborious annotations of pedestrians in every frame and camera view. Therefore, we propose in this paper an Unsupervised Multi-view Pedestrian Detection approach (UMPD) to learn an annotation-free detector via vision-language models and 2D-3D cross-modal mapping: 1) Firstly, Semantic-aware Iterative Segmentation (SIS) is proposed to extract unsupervised representations of multi-view images, which are converted into 2D masks as pseudo labels, via our proposed iterative PCA and zero-shot semantic classes from vision-language models; 2) Secondly, we propose Geometry-aware Volume-based Detector (GVD) to end-to-end encode multi-view 2D images into a 3D volume to predict voxel-wise density and color via 2D-to-3D geometric projection, trained by 3D-to-2D rendering losses with SIS pseudo labels; 3) Thirdly, for better detection results, i.e., the 3D density projected on Birds-Eye-View, we propose Vertical-aware BEV Regularization (VBR) to constrain pedestrians to be vertical like the natural poses. Extensive experiments on popular multi-view pedestrian detection benchmarks Wildtrack, Terrace, and MultiviewX, show that our proposed UMPD, as the first fully-unsupervised method to our best knowledge, performs competitively to the previous state-of-the-art supervised methods. Code is available at https://github.com/lmy98129/UMPD.

## CCS Concepts

• **Computing methodologies** → **Object detection**; **Unsupervised learning**; • **Information systems** → **Language models**.

## Keywords

Multi-view pedestrian detection, Unsupervised learning

---

*Corresponding author.

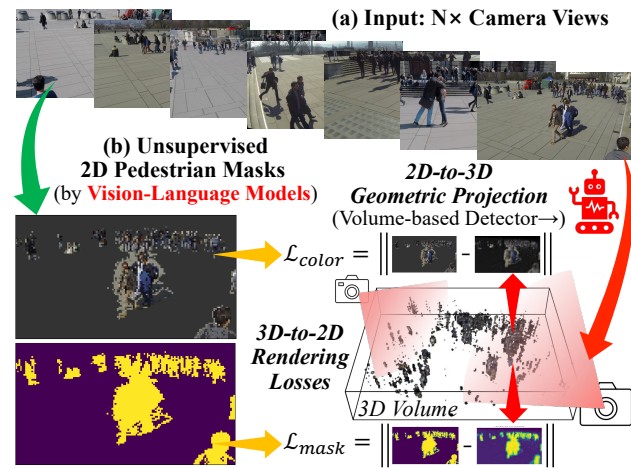

Figure 1: An overview of our proposed unsupervised pedestrian detection system. (a) The inputs are images from N× camera views. (b) Unsupervised 2D pedestrian masks are obtained via vision-language models. (c) With 2D-3D cross-modal mapping, 3D volume is predicted by 2D-to-3D geometric projection, learned from 3D-to-2D rendering losses, and projected on Birds-Eye-View (BEV) as detection results.

**ACM Reference Format:**
Mengyin Liu, Chao Zhu, Shiqi Ren, and Xu-Cheng Yin. 2024. Unsupervised Multi-view Pedestrian Detection. In *Proceedings of the 32nd ACM International Conference on Multimedia (MM '24), October 28-November 1, 2024, Melbourne, VIC, Australia.* ACM, New York, NY, USA, 9 pages. https://doi.org/10.1145/3664647.3681560

## 1 Introduction

Detecting pedestrians is fundamental in various real-world applications, especially where the fine-grained positions of pedestrians on Birds-Eye-View (BEV) are required rather than coarse-grained bounding boxes [21], such as crowd forecasting [1] for safety and customer behavior analysis [9] for retailing. To avoid occlusion or smaller scales, as shown in Figure 1(a), multiple cameras around an interested region are introduced to better detect pedestrians.

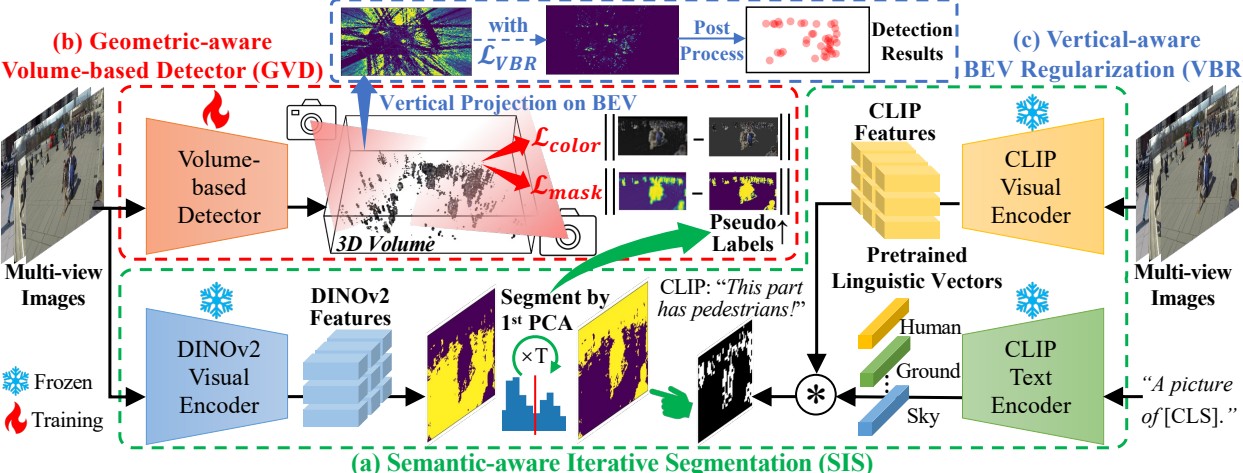

Figure 2: The architecture of our proposed UMPD approach. (a) Semantic-aware Iterative Segmentation (SIS) iteratively segments the PCA values of the DINOv2 [25] features into pseudo labels, based on vision-language model CLIP [28] as a foreground selector. (b) Geometric-aware Volume-based Detector (GVD) encodes multi-view 2D images into a 3D volume via 2D-3D cross-modal mapping, and learns to predict 3D density and color by $\mathcal{L}_{color}$ and $\mathcal{L}_{mask}$ with SIS pseudo labels (cameras mean rendering). (c) Vertical-aware BEV Regularization (VBR) constrains the predicted 3D density to be "human-like" vertical on BEV.

Most previous methods depend on supervised learning, including the classic detection-based [4, 36] and anchor-based detectors [2, 5], as well as more recent perspective-based anchor-free ones [7, 11, 12, 27, 31, 38]. They require supervised models to obtain 2D detection, segmentation or feature encoding, which are mapped onto the BEV for detection. Thus, pedestrian BEV positions are annotated.

However, heavy human labors are essential to annotate BEV labels of all-view video frames [4, 8]. Recently, with tinier pedestrians that harder to label, multi-view counting dataset [37, 39] to count coarse number is introduced to evaluate accurate BEV detection [38]. Although game engines can auto-generate images and labels [12, 39], domain gaps like different image styles or camera poses hinder better cross-domain performance [33] towards in-domain real scenes [4]. Therefore, manual labels are still inevitable.

As is shown in Figure 1(b) and (c), we found a potential solution without manual labels via 2D-3D cross-modal mapping. For 3D pedestrian density, the BEV labels mean a top-down observation, and 2D masks from cameras mean the surrounding observations. Hence, the 3D density becomes a "bridge", to be predicted by a volume-based detector from 2D images, learned from unsupervised 2D masks, and finally projected onto BEV as the detection results.

For 2D pedestrian masks, we notice the recent powerful unsupervised vision-language models like DINOv2 [25] and CLIP [28]. Robust representations with inter-image co-exist concepts can be extracted by DINOv2, and converted into the 1st Principal Component Analysis (PCA) values to be segmented as foreground masks. Meanwhile, CLIP model can identify object classes by texts, e.g., "A picture of a human", following the default template of CLIP [28].

To construct a 3D volume from multi-view images, some supervised methods for 3D object detection [30, 32] or 3D pose estimation [14] project each pixel of 2D features into potential 3D voxels based on geometric correspondence. Such powerful 3D volume frameworks inspire us to design a novel fully-unsupervised detector.

Furthermore, differentiable rendering framework [15] can render 3D density predicted by a volume-based detector into the 2D mask of each view, which is learned via unsupervised 2D masks. To better discriminate the pedestrian instances from their appearances, the colors are also rendered and learned via original 2D images.

In summary, we have observed a high dependency of the current mainstream supervised methods on the laborious manual labels. As is illustrated in Figure 2, we propose a novel approach to tackle this problem via **U**nsupervised **M**ulti-view **P**edestrian **D**etection (**UMPD**). Our main contributions are:

- Firstly, Semantic-aware Iterative Segmentation (SIS) method is proposed to extract the PCA values of DINOv2 representations, and segment them into 2D masks as pseudo labels. To identify the pedestrians, iterative PCA is adopted with zero-shot semantic classes of vision-language model CLIP.
- Secondly, we propose Geometric-aware Volume-based Detector (GVD) to encode multi-view 2D images into a 3D volume via geometry, and learn to predict 3D density and color from this volume via rendering losses with SIS pseudo labels.
- Thirdly, Vertical-aware BEV Regularization (VBR) method is further proposed to constrain the predicted 3D density to be vertical on BEV, following the natural pedestrian poses.
- Finally, formed by these key components, our proposed UMPD, **as the first fully-unsupervised method in this field to our best knowledge**, performs competitively on popular Wildtrack, Terrace, and MultiviewX datasets, especially compared with the previous supervised methods.

## 2 Related Works

### 2.1 Multi-view Pedestrian Detection

To detect the pedestrians in multi-view images, various detection methods have been proposed based on different architectures.

Following the pedestrian detection for monocular image, RCNN & Clustering [36] firstly detects pedestrians by a supervised 2D detector in each view, and then fuses them via clustering. Similarly, POM-CNN [4] fuses 2D masks by supervised model as final results. Due to weaker cross-view consistency of the single-view 2D boxes or masks, their performances are worse than BEV-based methods.

In an anchor-based style, DeepMCD [5] predicts the BEV positions by anchors. Deep-Occlusion [2] trains Conditional Random Field (CRF) on anchor features. In an anchor-free style, MVDet [12] firstly uses homography mapping from 2D features onto BEV to predict positions. Following MVDet, SHOT [31] makes multi-height projection. 3DROM [27] learns with a random occlusion augmentation. More augmentation is adopted by MVAug [7]. MVDeTr [11] predicts pedestrian directions via Transformer. With source domain labels, GMVD [33] is still inferior to in-domain methods. Recently, an unsupervised view fusion component is applied to supervised detection [38]. All these methods rely on laborious BEV labels.

Differently, without any manual labels, our proposed volume-based detector GVD performs 2D-3D cross-modal mapping to predicting the 3D density of pedestrians, learned from our 2D SIS masks as pseudo labels with our VBR as an extra regularization.

## 2.2 Unsupervised Feature Representation

In the past decade, unsupervised learning of feature representations has achieved powerful performance. Specifically, popular contrastive learning discriminates paired samples, which is capable of more unsupervised zero-shot tasks than supervised fine-tuning.

For single modality, DINOv1 [3] learns a self-distillation for only positive samples, which is utilized by CutLER [34] to segment all salient objects in a single 2D image rather than multi-view ones. Recently, DINOv2 [25] with better data quality and learning scheme yields more robust representations, which segments shared concepts across images, by dividing the 1$^{st}$ PCA values of features.

For multiple modalities, CLIP [28] learns on paired images and texts collected from the internet. Furthermore, MaskCLIP [40] predicts 2D masks queried by input texts, and CrowdCLIP [20] merely predicts people count. But these unsupervised methods are too coarse-grained for accurate masks as single-modal DINOs [3, 25].

In this paper, we complement these powerful unsupervised models as our proposed SIS to obtain pseudo labels for Unsupervised Multi-view Pedestrian Detection (UMPD), which iteratively segments PCA of DINOv2 features for 2D masks, with zero-shot semantic capability of vision-language model CLIP to identify pedestrians.

## 2.3 Multi-view Construction of 3D Volume

For various 3D perception tasks, it is crucial to construct an explicit 3D volume from multi-view images. For example, 3D pose estimation methods [14] or 3D object detectors [30, 32] assigns 2D features to their corresponding 3D voxels as an explicit volume.

Neural Radiance Field (NeRF) [17, 23, 24] learns a neural network to represent implicit volume. However, typical $\leq 10$ [4, 12] or even 4 views [8] in multi-view pedestrian detection datasets are too sparse for RFP [23] that needs 20~80 views, and LERF [17] that fits videos with 400~600 views. They distill implicit semantics of DINO [3] and CLIP [28], while our SIS yields explicit 2D labels by [25, 28]. Implicit volume is also inapt for vertical constraints like our VBR.

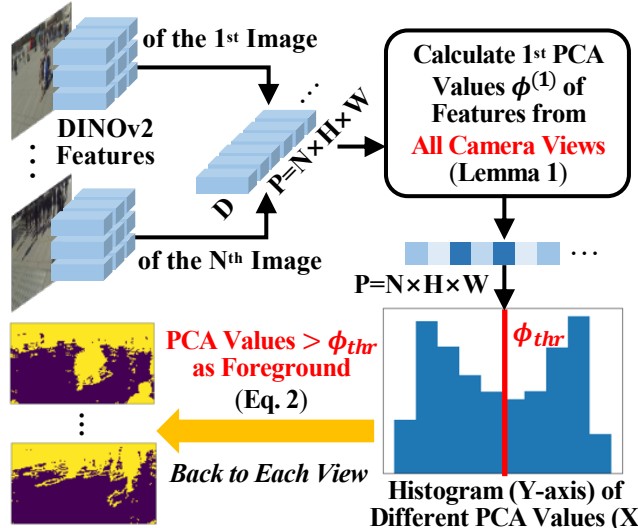

**Figure 3: Our proposed unsupervised segmentation of the 1$^{st}$ PCA. DINOv2 features $X \in \mathbb{R}^{P \times D}$ represent distinguished background and "foreground" across all views, e.g., BEV ground plane *vs* pedestrians and their contexts, which are mapped into the 1$^{st}$ PCA values $\Phi \in \mathbb{R}^P$ for a further division.**

Another explicit technique is unsupervised Multi-View Stereo (MVS) [19, 35], but 3D volume of the whole scene is constructed from images, rather than only foreground like pedestrians. Differently, for 3D volume by our proposed GVD, differentiable rendering framework PyTorch3D [15] renders 3D densities and colors into both 2D masks and images, learned via our SIS pedestrian masks.

## 3 Proposed Method

As is illustrated in Figure 2, our proposed UMPD approach comprises three key components: 1) Semantic-aware Iterative Segmentation (SIS) segments the PCA values into masks iteratively, with zero-shot semantic capability of CLIP to identify pedestrians; 2) Geometric-aware Volume-based Detector (GVD) encodes multi-view 2D images into a 3D volume via geometric correspondence, and learns to predict density and color by the rendering losses with SIS masks; 3) Vertical-aware BEV Regularization (VBR) method constrains the 3D density from GVD to be vertical on the BEV plane. More details will be introduced in the following sections.

## 3.1 Semantic-aware Iterative Segmentation

With the powerful unsupervised methods like DINOv2 [25], similar PCA values of cross-image features indicate the same concepts, which can be better controlled than the merely single-image salient objects in the previous unsupervised methods [3, 34]. In the figures of DINOv2 paper, even abstract concepts like "wings" of airplane and bird in different images are co-segmented. Thus, it is adopted to identify pedestrians based on multi-view cross-image information.

*3.1.1 Unsupervised Segmentation of the 1$^{st}$ PCA.* As shown in Figure 3, DINOv2 features of N× image views are $X = (X_1^\top, X_2^\top, \cdots, X_P^\top)^\top$, where $P = N \times H \times W$ is total pixels and each $X_p \in \mathbb{R}^D$. Denote

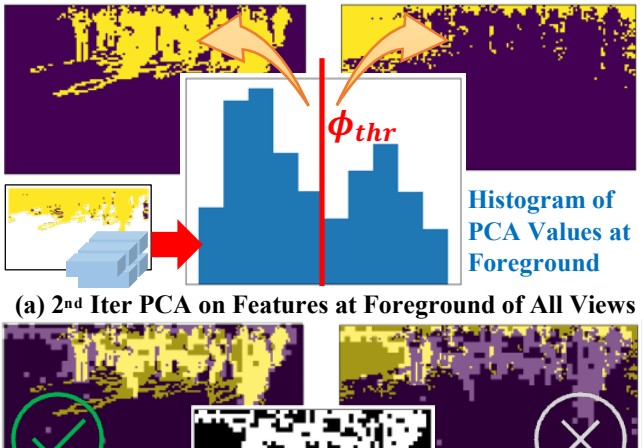

(a) 2$^{nd}$ Iter PCA on Features at Foreground of All Views

"More overlap. =Foreground!"   From CLIP (Eq. 3)   "Less overlap. =Background."

(b) CLIP to Identify the Foreground after 2$^{nd}$ Iter PCA

**Figure 4: Our proposed Semantic-aware Iterative Segmentation (SIS). (a) Since the 1$^{st}$ iteration of PCA yields coarse-grained "foreground" with pedestrians and their contexts, the 2$^{nd}$ iteration is performed, with PCA values and their histograms in XY-axes. (b) Vision-language model CLIP [28] identifies the foreground after the 2$^{nd}$ iteration of PCA.**

$Z^{(1)}$ as a linear mapping from $\mathbf{X}$ to $\phi^{(1)} = (\phi_1^{(1)}, \phi_2^{(1)}, \cdots, \phi_P^{(1)})^\top$, where $\sum_{p=1}^{P} x_{pd} = 0$ after the Zero Standardization of each $X_p$:

$$Z^{(1)} = \phi_1^{(1)} X_1 + \phi_2^{(1)} X_2 + \cdots + \phi_P^{(1)} X_P. \quad (1)$$

LEMMA 1. *The 1$^{st}$ PCA vector $\phi^{(1)}$ of $\mathbf{X}$: (a) maximizes the global variance $\mathrm{var}(\phi^{(1)}) = \frac{1}{P} \sum_{p=1}^{P} (\phi_p^{(1)})^2$; (b) minimizes the reconstruction loss $\min_{Z^{(1)}} \|\mathbf{X} - \mathbf{X} Z^{(1)} (Z^{(1)})^\top\|^2$, where $Z^{(1)}$ is a bi-direction mapping between high-dimensional $\mathbf{X}$ and low-dimensional $\phi^{(1)}$.*

Given the foreground or background that co-exist in multiple images, if their DINOv2 features $\mathbf{X}$ are similar, according to the bi-direction mapping ensured by a minimized reconstruction loss in Lemma 1, the 1$^{st}$ PCA values $\phi^{(1)}$ of these features are also similar and distinguished from the opposite parts, as is illustrated by the two peaks in the histogram of Figure 3. Then, $\phi_p^{(1)} \in \mathbb{R}$ as scalars can be more easily divided by a threshold value than the complicated DINOv2 feature vectors $X_p \in \mathbb{R}^D$. The proofs of Lemma 1 are provided in our supplementary materials†.

Therefore, given the pixel-wise PCA values $\Phi_{nij}$ of $\Phi \in \mathbb{R}^{N \times H \times W}$, where $\Phi$ are reshaped from the vector $\phi^{(1)} \in \mathbb{R}^P$, they are segmented into 2D masks $\mathbf{M}_n^{t=1}$ of each view $n$ in Figure 3 via the threshold value $\phi_{thr}$, which is formulated as:

$$\mathbf{M}_{nij}^{t=1} = \begin{cases} 1.0, & \Phi_{nij} > \phi_{thr}, \text{ as foreground,} \\ 0.0, & \Phi_{nij} \leq \phi_{thr}, \text{ as background.} \end{cases} \quad (2)$$

† For further proofs, experimental results, and visualizations, please refer to our supplementary materials: https://lmy98129.github.io/academic/src/UMPD-Appendix.pdf.

---

**Algorithm 1** Semantic-aware Iterative Segmentation

1: **function** SEMITERSEG($\mathbf{I}$, $T_{PCA}$, UseCLIP)
2:     Prompts ← 'A picture of' + {'human', 'ground', 'sky'}
3:     $\mathbf{X} \leftarrow \varnothing; \mathbf{S} \leftarrow \varnothing$
4:     **for** $n \in [1, N]$ **do**       ▷ Features & CLIP masks per view $n$
5:         $\mathbf{X} \leftarrow \mathbf{X} \cup \text{DINOv2}(\mathbf{I}^n); \mathbf{S} \leftarrow \mathbf{S} \cup \text{CLIP}(\mathbf{I}^n, \textbf{Prompts})$
6:
7:     $\mathbf{X}^1 \leftarrow \mathbf{X}$                ▷ Feature collection of all views
8:     **for** $t \in [1, T_{PCA}]$ **do**      ▷ Iterative Segmentation of PCA
9:         $\Phi^t = \text{PCA}(\mathbf{X}^t)$                          ▷ Lemma 1
10:         $\mathbf{M}^t \leftarrow$ Filter PCA values $\Phi^t$ by Eq. 2
11:         **if** UseCLIP **then**
12:             $\mathbf{M}^t \leftarrow$ Select foreground by $\mathbf{S}$ of CLIP in Eq. 3
13:         **if** t>1 **then**              ▷ Update foreground
14:             $\mathbf{M}^{t-1}[\mathbb{1}(\mathbf{M}^{t-1} = 1.0)] \leftarrow \mathbf{M}^t; \mathbf{M}^t \leftarrow \mathbf{M}^{t-1}$
15:         $\mathbf{X}^{t+1} \leftarrow \mathbf{X}^t[\mathbb{1}(\mathbf{M}^t = 1.0)]$        ▷ Update features
16:     **return** $\mathbf{M}^{T_{PCA}}$             ▷ Output pseudo labels

---

*3.1.2 Semantic-aware Iterative Segmentation of PCA.* Recalling the Lemma 1, DINOv2 features are bi-directionally mapped to their PCA values. However, as is illustrated in the Figure 3, they represent the ground plane as the most distinguished background, but the remaining parts are pedestrians and their contexts. Thus, more PCA iterations in Algorithm 1 is used to further segment the pedestrians and non-human background inside these coarse-grained features.

Firstly, "foreground" features $\mathbf{X}^t[\mathbb{1}(\mathbf{M}^t = 1.0)] \in \mathbb{R}^{P' \times D}, P' \ll P$ are collected to calculate the 1$^{st}$ PCA values and yield new mask $\mathbf{M}^t$ by Eq.2. Differently, the mask is only updated inside foreground of $t-1$, where new background is thus merged into the previous ones. However, in Figure 4(a), the $\leq \phi_{thr}$ part is the real foreground, which is inconsistent with Eq. 2, because merely inter-image concepts are segmented, regardless of which part means the real pedestrians.

Inspired by the zero-shot semantic capability to recognize the object classes [20] of vision-language models like CLIP [28], the original pooling is modified [40] to obtain 2D masks. Denoted as "⊛" in Figure 2(a), cosine similarity $S_{nij}^c = ((\mathcal{W}^c)^\top (\mathcal{V}_{nij}))/(\|\mathcal{W}^c\| \|\mathcal{V}_{nij}\|)$ is calculated between pretrained linguistic vector $\mathcal{W}^c \in \mathbb{R}^{D'}$ of each class $c$ and the vision feature $\mathcal{V}_{nij} \in \mathbb{R}^{D'}$ of each pixel $(i, j)$ of view $n$ to indicate semantic classes (i.e., "human", "ground", and "sky", background classes are necessary), where $\mathbf{S}^{human} \in \mathbb{R}^{N \times H \times W}$. Given a view $n$ like Figure 4(b), foreground $\mathbf{M}_n^t$ is decided by the overlapping "∩" with $\mathbf{S}_n^{human} \in \mathbf{S}$:

$$\mathbf{M}_n^t = \begin{cases} \mathbf{M}_n^t, & \text{if} \|\mathbf{S}_n^{human} \cap \mathbf{M}_n^t\| > \|\mathbf{S}_n^{human} \cap (1 - \mathbf{M}_n^t)\|, \\ 1 - \mathbf{M}_n^t, & \text{otherwise.} \end{cases} \quad (3)$$

## 3.2 Geometric-aware Volume-based Detector

Manual BEV labels and 2D pedestrian masks from SIS can represent different observations of 3D pedestrians, i.e., the former are the top-down observations, and the latter are the surrounding ones.

To learn from these pseudo labels and predict the BEV pedestrian positions, a 3D volume as a "bridge" between them is constructed from encoding the input multi-view 2D images by our proposed fully-unsupervised detector, which is a 2D-3D cross modal mapping.

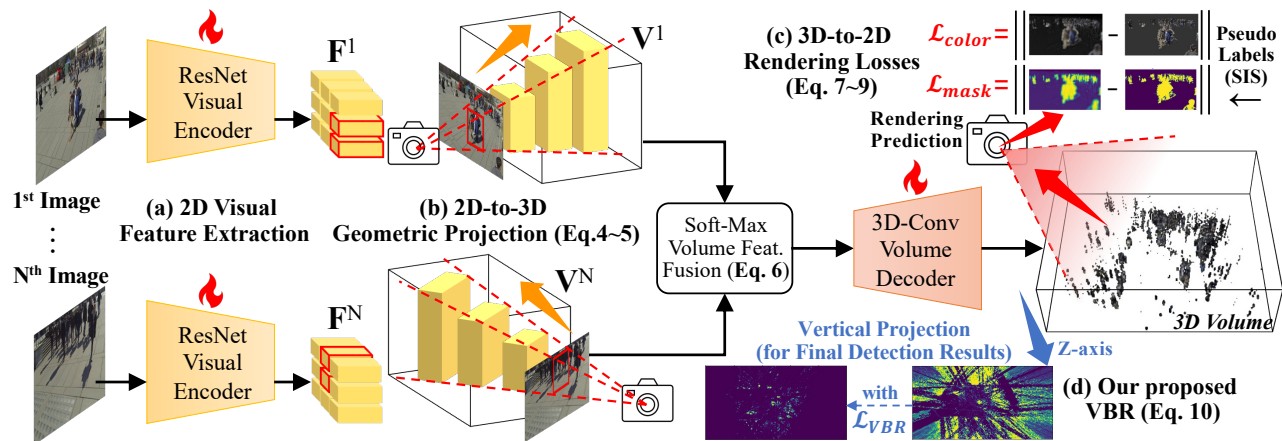

**Figure 5: Our proposed Geometric-aware Volume-based Detector (GVD) and Vertical-aware BEV Regularization (VBR). (a) 2D features $F^n$ of each view $n$ are extracted by a visual encoder. (b) 2D-to-3D geometric projection assigns 2D features (like highlighted red boxes) to potential 3D voxels. (c) 3D-to-2D rendering losses with SIS labels render the predicted 3D density and color by PyTorch3D [15] to 2D masks and images. (d) Our proposed VBR constrains vertical density for better detection results.**

*3.2.1  2D-to-3D Geometric Projection for 3D Volume.* Inspired by the previous multi-view 3D methods [30, 32] for supervised tasks, we extract 2D features $\mathbf{F}^n \in \mathbb{R}^{C \times H \times W}$ of each view $n$ via a visual encoder [10] as is shown in Figure 5(a), where C is the channel number. Then, pixel-wise feature $F_{uv}^n \in \mathbb{R}^C$ is back-projected into potential 3D voxels via the geometry of 2D-3D cross modal mapping.

Formally, given a pinhole camera calibrated with intrinsic and extrinsic matrices $\{\mathbf{K}^n, \mathbf{\Psi}^n\}$, its geometric model to obtain a 2D pixel $(u, v)^\top$ captured from a 3D voxel position $(x, y, z)^\top$ is:

$$\begin{bmatrix} u \\ v \\ 1 \end{bmatrix} = \frac{1}{\lambda} \mathbf{K}^n \mathbf{\Pi}^0 \mathbf{\Psi}^n \begin{bmatrix} x \\ y \\ z \\ 1 \end{bmatrix}, \mathbf{\Pi}^0 = \begin{bmatrix} 1 & 0 & 0 & 0 \\ 0 & 1 & 0 & 0 \\ 0 & 0 & 1 & 0 \end{bmatrix}, \quad (4)$$

where $\mathbf{\Pi}^0$ is an auxiliary matrix to obtain $(x', y', z')^\top$ from the 3D homogeneous coordinate $(x', y', z', 1)^\top$ after the transformation of extrinsic matrix $\mathbf{\Psi}^n$, and $\lambda$ is the depth distance between this 3D voxel and camera optical origin. $\frac{1}{\lambda}$ ensures the 2D homogeneity of $(u, v, 1)^\top$ from $(u', v', \lambda)^\top$ after applying intrinsic matrix $\mathbf{K}$.

In 3D geometry, there are various 3D voxels $(x, y, z)^\top$ with different potential depths $\lambda$ in Eq. 4, that derive the same 2D pixel $(u, v, 1)^\top$. For such a one-to-many 2D-3D cross-modal correspondence, the 2D pixel feature $F_{uv}^n$ is assigned to all these 3D voxel positions to form a 3D volume $\mathbf{V}^n \in \mathbb{R}^{C \times X \times Y \times Z}$ from each view $n$:

$$\mathbf{V}^n[:, x, y, z] = \mathbf{F}^n[:, u, v]. \quad (5)$$

To fuse the 3D volumes $\mathbf{V}^n$ of each view $n$ in Figure 5(b) into a unified $\mathbf{V}$, a soft-max function is used like supervised volume-based methods [14] to re-weight $\mathbf{V}^n$, where the highest feature values at its voxel in all the views $n$ are assigned with the largest weights:

$$\mathbf{V} = \sum_n \left( \frac{\exp(\mathbf{V}^n)}{\sum_n \exp(\mathbf{V}^n)} \circ \mathbf{V}^n \right), \quad (6)$$

where $\mathbf{V} \in \mathbb{R}^{C \times X \times Y \times Z}$ and $\circ$ denotes Hadamard Product. To predict the 3D density $\mathcal{D} \in [0, 1]^{X \times Y \times Z}$ and color $C \in [0, 1]^{3 \times X \times Y \times Z}$ from the volume $\mathbf{V}$, 3D-convolution with a $5 \times 5 \times 5$ kernel is adopted

as a decoder for a larger 3D receptive field, like the large-kernel 2D-convolution of the supervised methods [12] on the BEV features.

*3.2.2  3D-to-2D Rendering Losses with Pseudo Labels.* With the 2D pedestrian masks $\mathbf{M}$ as pseudo labels, differential rendering framework PyTorch3D [15] renders the predicted 3D density $\mathcal{D}$ into 2D masks $\tilde{\mathbf{M}}$ by the pinhole camera model in Eq. 4.

In details, given a 2D pixel position $(u, v)^\top$ in a view $n$, the 3D direction of the ray $r$ from the 3D position of camera optical origin $o$ through $(u, v)^\top$ is $\delta$. Any 3D position along this ray is $r(\tau) = o + \tau(\delta)$. $\tau$ is the distance between $r(\tau)$ and $o$, whose maximum is $\Lambda$. The rendered 2D mask pixel $\tilde{\mathbf{M}}^n(u, v) = \tilde{\mathbf{M}}(\tau)$ from $\mathcal{D}$ is denoted as:

$$\tilde{\mathbf{M}}(\tau) = \int_0^\Lambda \mathcal{T}(\tau) \mathcal{D}(r(\tau)) d\tau, \mathcal{T}(\tau) = \exp\left(-\int_0^\tau \mathcal{D}(r(s)) ds\right), \quad (7)$$

where $\mathcal{T}(\tau)$ is the transmittance (i.e., optical opacity) at distance $\tau$.

Similarly, to discriminate crowded pedestrians by their appearances, the predicted 3D colors $C$ are also rendered into 2D images $\tilde{\mathbf{I}}$. Each 2D pixel $\tilde{\mathbf{I}}^n(u, v) = \tilde{\mathbf{I}}(\tau)$ of rendered image is formulated as:

$$\tilde{\mathbf{I}}(\tau) = \int_0^\Lambda \mathcal{T}(\tau) \mathcal{D}(r(\tau)) C(r(\tau)) d\tau. \quad (8)$$

Note that predicted 3D color $C$ is view-invariant like the real world, i.e., it is identical in any observations, thus input is 3D position $r(\tau)$ without view direction $\delta$ as NeRF [24]. Since only foreground have colors, the pseudo labels are images $\mathbf{I}$ masked by $\mathbf{M}$.

Following the official instruction of PyTorch3D [15], Huber Loss $\mathcal{L}_{huber}$ [13] is used with inputs $\tilde{\theta}_{nij} \in \tilde{\Theta}$, $\theta_{nij} \in \Theta$ as 2D predictions and pseudo labels for 2D mask $\mathbf{M}$ or image $\mathbf{I}$ in Figure 5(c):

$$\mathcal{L}_{huber}(\tilde{\Theta}, \Theta) = \frac{1}{NHW} \sum_{n,i,j=1}^{NHW} \left| \sqrt{\|\tilde{\theta}_{ij}^n - \theta_{ij}^n\|^2 + 1} - 1 \right|. \quad (9)$$

## 3.3  Vertical-aware BEV Regularization

As is shown in Figure 2(c) and 5(d), if predicted pedestrians are leaning or laying down, their BEV occupancies, i.e., maximized 3D

**Table 1: Detailed information of the multi-view pedestrian detection datasets for performance evaluation.**

| Datasets | Camera Number | Input Resolution | Data Collection | Train Frames | Test Frames | Area $(m \times m)$ | Crowdedness (person/frame) |
|---|---|---|---|---|---|---|---|
| Wildtrack [4] | 7 | $1920 \times 1080$ | Real World | 360 | 40 | $12 \times 36$ | 20 |
| Terrace [8] | 4 | $360 \times 288$ | Real World | 300 | 200 | $5.3 \times 5$ | 10 |
| MultiviewX [12] | 6 | $1920 \times 1080$ | Simulation | 360 | 40 | $16 \times 25$ | 40 |

**Table 2: Ablation study on our proposed Semantic-aware Iterative Segmentation (SIS) about the PCA iteration $T_{PCA}$ and the usage of zero-shot semantic capability from unsupervised vision-language model CLIP to identify pedestrians.**

| $T_{PCA}$ | UseCLIP | MODA | MODP | Precision | Recall |
|---|---|---|---|---|---|
| 1 | | 15.0 | 58.0 | 95.0 | 15.9 |
| | ✓ | 19.1 | 55.1 | 68.9 | 34.9 |
| 2 | | 49.8 | 58.5 | 89.5 | 56.4 |
| | ✓ | **76.6** | **61.2** | 90.1 | 86.0 |
| 3 | | 9.1 | 43.7 | 85.0 | 11.0 |
| | ✓ | 57.4 | 60.9 | 91.4 | 63.3 |

**Table 3: Ablation study on the threshold $\phi_{thr}$ of our proposed SIS, where $\phi_{thr} = 2$ is the overall best among all benchmarks.**

| $\phi_{thr}$ | Wildtrack | Terrace | MultiviewX | Average Rank |
|---|---|---|---|---|
| 1 | 61.1 | 57.9 | 78.9 | 2.7 |
| **2** | **61.2** | 59.0 | **79.4** | **1.6** |
| 3 | 60.9 | **59.7** | 77.6 | 2.3 |

**Table 4: Ablation study on our proposed Geometric-aware Volume-based Detector (GVD) and Vertical-aware BEV Regularization (VBR). Concatenation and adding are compared with Eq. 6. Loss functions $\mathcal{L}_{color}$ and $\mathcal{L}_{VBR}$ are also ablated.**

| Settings | MODA | MODP | Precision | Recall |
|---|---|---|---|---|
| **our UMPD** | **76.6** | **61.2** | 90.1 | 86.0 |
| Eq. 6 → Concat. | 71.3 | 60.8 | 83.4 | 89.1 |
| Eq. 6 → Adding | 74.5 | 61.0 | 86.1 | 88.9 |
| → w/o $\mathcal{L}_{color}$ | 71.0 | 60.3 | 83.2 | 89.0 |
| → w/o $\mathcal{L}_{VBR}$ | 27.3 | 52.1 | 65.9 | 56.7 |

density along Z-axis, are unnaturally larger than standing. Thus, Vertical-aware BEV Regularization (VBR) is proposed to regularize the pedestrian poses with minimized BEV occupancy:

$$\mathcal{L}_{VBR}(\mathcal{D}) = \frac{1}{XY} \sum_{x,y=1}^{XY} \left| \max(\mathcal{D}_{xyz})_z \right|, \mathcal{D}_{xyz} \in \mathcal{D}. \quad (10)$$

Finally, the overall loss function $\mathcal{L}$ that optimizes the parameters $\theta^* = \arg\min \mathcal{L}$ of our UMPD to predict the best 3D color and density based on rendered $\tilde{\mathbf{I}}$ and $\tilde{\mathbf{M}}$ in Eq. 8 and 7 is formulated as:

$$\begin{aligned}
\mathcal{L} &= \mathcal{L}_{color} + \mathcal{L}_{mask} + \mathcal{L}_{VBR} \\
&= \mathcal{L}_{huber}(\tilde{\mathbf{I}}, \mathbf{I}) + \mathcal{L}_{huber}(\tilde{\mathbf{M}}, \mathbf{M}) + \mathcal{L}_{VBR}(\mathcal{D}),
\end{aligned} \quad (11)$$

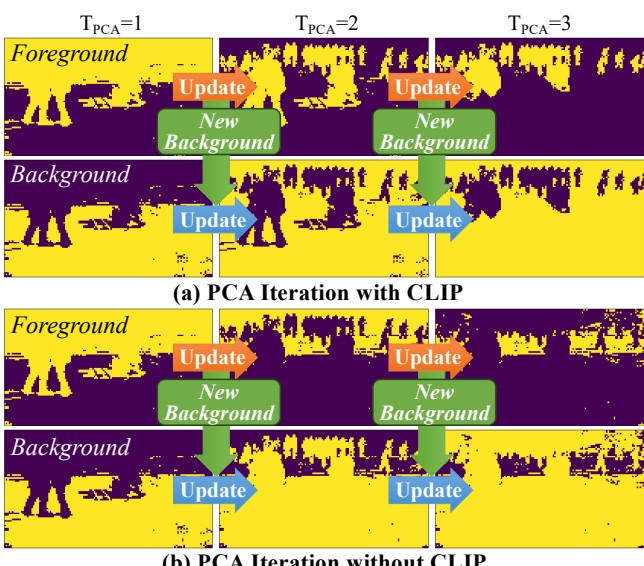

(a) PCA Iteration with CLIP

(b) PCA Iteration without CLIP

**Figure 6: Visualization of different PCA iteration $T_{PCA}$ and CLIP usage in our proposed SIS. Orange and green arrows are foreground and new background updated from previous foreground. Blue arrows are the inherited previous background.**

where weights = 1.0 of all loss item follow popular supervised multi-view 3D detectors [30, 32]. For inference, the detection results of our proposed UMPD are the predicted 3D density $\mathcal{D}$ projected on BEV, i.e., $\Omega = \max(\mathcal{D}_{xyz})_z \in \mathbb{R}^{X \times Y}$, where $\mathcal{D}_{xyz} \in \mathcal{D}$.

## 4 Experiments

In this section, extensive experiments are conducted on popular multi-view pedestrian detection benchmarks, i.e., Wildtrack, Terrace, and MultiviewX, to evaluate our proposed UMPD. Ablation study is performed on our proposed key components SIS, GVD, and VBR. Furthermore, qualitative analysis and state-of-the-art comparisons on these benchmarks are also reported.

### 4.1 Datasets

Wildtrack [4] and Terrace [8] datasets are collected from real-world interested regions surrounded by multiple calibrated and synchronized cameras, where the unscripted pedestrians are naturally walking or standing. MultiviewX [12] is a challenging dataset using Unity 3D Engine to simulate more populated scenes, featured with higher crowdedness than [4, 8]. Multiple Object Detection Accuracy and Precision [16] (MODA and MODP), Precision, and Recall are evaluation metrics. Table 1 shows more details of these datasets.

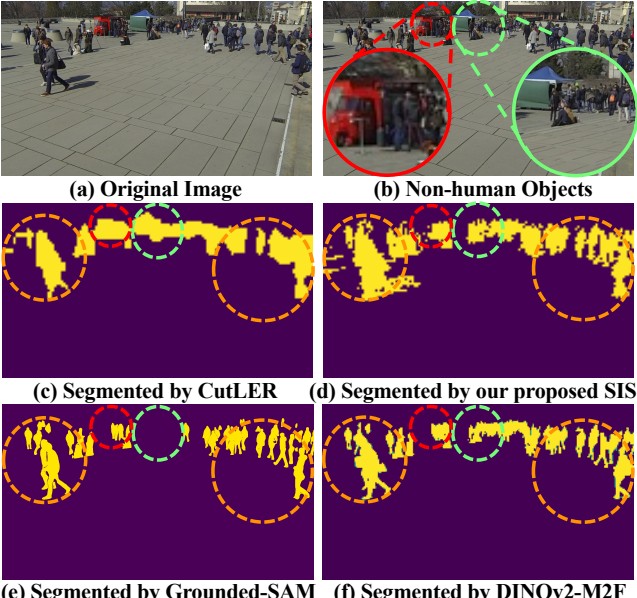

**(a) Original Image**       **(b) Non-human Objects**

**(c) Segmented by CutLER**  **(d) Segmented by our proposed SIS**

**(e) Segmented by Grounded-SAM**   **(f) Segmented by DINOv2-M2F**

**Figure 7: Different 2D mask segmentation results. CutLER [34] segments salient objects in red and green circles, while our multi-view SIS avoids them. Grounded-SAM [29] ignores small pedestrians in green circle. Supervised DINOv2-based Mask2Former [6] yields fine-grained masks in orange circles.**

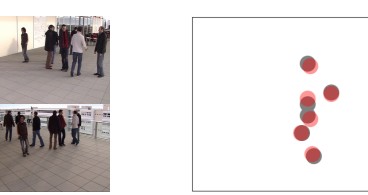

**Figure 8: Detection results of UMPD on Terrace [8] dataset.**

## 4.2 Implementation Details

Our proposed UMPD method is based on PyTorch [26] framework. For our SIS, unsupervised features for PCA with $\phi_{thr} = 2$ are extracted by ViT-B/14 DINOv2 model [25]. $S^{human}$ is generated by ResNet-50 [10] of unsupervised vision-language model CLIP [28, 40]. The visual encoder of our GVD is ResNet-18 [10] following [12]. The result $\Omega$ is post-processed with a 3×3 Gaussian Kernel and > 0.4 threshold used by [12]. 4×A5000 GPUs are used for training with a mini-batch 4 and learning rate $1 \times 10^{-2}$, and 1×GPU for testing. Our SIS and VBR are only for training. The inference time of GVD is ~1.0s/frame. Each frame comprises multi-view images.

## 4.3 Ablation Study

The ablation study is performed on the popular real-world dataset Wildtrack. In Table 2, for our proposed SIS, different PCA iteration $T_{PCA}$ and the zero-shot semantic capability of CLIP are experimented. "$T_{PCA}=1$" performs worse with coarse-grained foreground as shown in Figure 3 Meanwhile, "$T_{PCA}=3$" causes over-segmentation. Although CLIP fixes some mistakes by keeping some

**Table 5: Different 2D mask segmentation methods for pseudo labels: our proposed SIS, single-image method CutLER [34], supervised Mask2Former [6] based on DINOv2 [25], and Grounded-SAM [29] with Grounding-DINO [22] prompts.**

| Methods | MODA | MODP | Precision | Recall |
|---|---|---|---|---|
| SIS(=**UMPD**) *DINOv2,unsp* | 76.6 | 61.2 | 90.1 | 86.0 |
| → CutLER *DINOv1,unsp* | 38.9 | 53.5 | 80.5 | 51.3 |
| → DINOv2-M2F *DINOv2,supv* | 79.3 | 63.3 | 90.1 | 89.1 |
| → Grnded-SAM *SAM+GD,supv* | 60.1 | 58.1 | 92.3 | 65.5 |

pedestrians parts as the foregrounds, the results are still not ideal. Finally, our SIS is equipped with CLIP to identify pedestrians and "$T_{PCA}=2$" as a proper setting. For the threshold $\phi_{thr}$ of our SIS, the overall best $\phi_{thr} = 2$ is obtained among all datasets in Table 3.

Meanwhile, Table 4 evaluates our proposed GVD and VBR by: 1) 2D-to-3D geometric projection with different volume fusion; 2) 3D-to-2D rendering losses such as $\mathcal{L}_{color}$ and $\mathcal{L}_{VBR}$, since $\mathcal{L}_{mask}$ is inevitable to predict detection results $\Omega$. For the volume fusion operations, the soft-max re-weighting in Eq. 6 achieves higher MODA and MODP, while simpler adding and concatenation are insufficient to directly handle such complex 3D volumes, which is consistent with the experiments on other multi-view tasks [14].

For the loss functions, $\mathcal{L}_{color}$ assists to detect better by discriminating the different appearances of pedestrian instances than merely $\mathcal{L}_{mask}$. Note that our $\mathcal{L}_{VBR}$ follows the natural vertical human poses and thus brings significant improvements (+49.3 MODA and +9.1 MODP) on the default $\mathcal{L}_{mask}$ and $\mathcal{L}_{color}$ following PyTorch3D [15].

## 4.4 Different 2D Mask Segmentation Methods

As is discussed in Section 2.2, CutLER [34] based on DINOv1 [3] is different from our proposed SIS with DINOv2 [25], which segment all salient objects in a single image, rather than inter-image concepts. Released by the DINOv2 official code [25], a new version of Mask2Former [6] is initialized with DINOv2 pre-training, and fine-tuned with supervision. Recently, Grounded-SAM [29] embraces SAM [18] with boxes from Grounding-DINO [22] as prompts.

In Table 5, we compare different 2D masks from these methods, as pseudo labels to train our UMPD. Single-view masks from CutLER without cross-view information performs worse. Similarly, single-view object boxes by Grounding-DINO [22] mislead Grounded-SAM [29] to yield low-quality masks. Mask2Former achieves better results by powerful pretraining and supervised fine-tuning, while our proposed fully-unsupervised SIS also performs competitively. In summary, the quality of 2D masks greatly affects the performance of our UMPD, which is worth future researches for better masks.

## 4.5 Qualitative Analysis

In Figure 6, we visualize the segmentation results of our unsupervised SIS with different PCA iteration $T_{PCA}$ and whether to use the vision-language model CLIP. At $T_{PCA}=1$, foregrounds in Figure 6(a) and (b) are coarse-grained, which is handled by > $\phi_{thr}$ part in Eq.

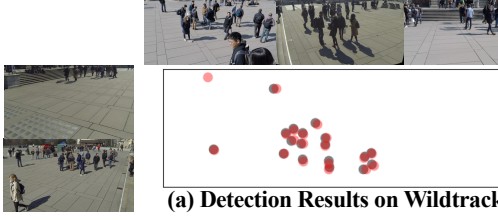
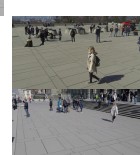
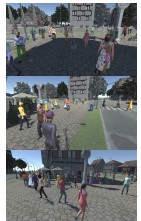
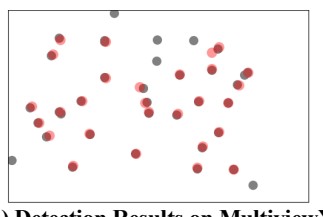
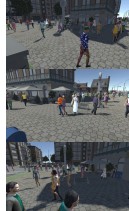

(a) Detection Results on Wildtrack

(b) Detection Results on MultiviewX

Figure 9: Detection results by our proposed UMPD on Wildtrack [4] and MultiviewX [12] datasets. Gray and Red dots are ground truths and the detection results from our UMPD, respectively. (a) Grouped crowded pedestrians on Wildtrack are correctly detected by our UMPD. (b) MultiviewX brings more challenges by larger-scale and more populated scenes than real-world ones.

Table 6: Comparisons with the state-of-the-arts on multi-view pedestrian detection datasets. Our UMPD is fully-unsupervised.

| Methods | Wildtrack | | | | Terrace | | | | MultiviewX | | | |
|---|---|---|---|---|---|---|---|---|---|---|---|---|
| | MODA | MODP | Precision | Recall | MODA | MODP | Precision | Recall | MODA | MODP | Precision | Recall |
| RCNN & Clus. | 11.3 | 18.4 | 68 | 43 | -11 | 28 | 39 | 50 | 18.7 | 46.4 | 63.5 | 43.9 |
| POM-CNN | 23.2 | 30.5 | 75 | 55 | 58 | 46 | 80 | 78 | - | - | - | - |
| DeepMCD | 67.8 | 64.2 | 85 | 82 | - | - | - | - | 70.0 | 73.0 | 85.7 | 83.3 |
| Deep-Occ. | 74.1 | 53.8 | 95 | 80 | 71 | 48 | 88 | 82 | 75.2 | 54.7 | 97.8 | 80.2 |
| MVDet | 88.2 | 75.7 | 94.7 | 93.6 | 87.2 | 70.0 | 98.2 | 88.8 | 83.9 | 79.6 | 96.8 | 86.7 |
| SHOT | 90.2 | 76.5 | 96.1 | 94.0 | 87.1 | 70.3 | 98.9 | 88.1 | 88.3 | 82.0 | 96.6 | 91.5 |
| MVDeTr | 91.5 | 82.1 | 97.4 | 94.0 | - | - | - | - | 93.7 | 91.3 | 99.5 | 94.2 |
| 3DROM | 93.5 | 75.9 | 97.2 | 96.2 | 94.8 | 70.5 | 99.7 | 95.1 | 95.0 | 84.9 | 99.0 | 96.1 |
| MVAug | 93.2 | 79.8 | 96.3 | 97.0 | - | - | - | - | 95.3 | 89.7 | 99.4 | 95.9 |
| GMVD | 70.7‡ | 73.8‡ | 89.1‡ | 80.6‡ | - | - | - | - | 88.2 | 79.9 | 96.8 | 91.2 |
| **UMPD (ours)** | 76.6 | 61.2 | 90.1 | 86.0 | 73.8 | 59.0 | 88.6 | 84.8 | 67.5 | 79.4 | 93.4 | 72.6 |

2. At $T_{PCA}$=2, the foreground is corrected by CLIP in Figure 6(a), while $> \phi_{thr}$ in Eq. 2 is used in Figure 6(b) and yields wrong mask, since PCA performs naïve segmentation regardless of semantic classes. Under the over-segmentation at $T_{PCA}$=3, CLIP decreases some mistakes, which is consistent with the results in Table 2.

In Figure 7, 2D masks of different segmentation methods are visualized. Noisy single-view masks by CutLER [34] comprise non-human salient objects like tent and truck. Wrong boxes without small pedestrians from Grounding-DINO [22] mislead the Grounded-SAM [29]. With DINOv2 pre-training and supervised fine-tuning, Mask2Former [6] performs better than our unsupervised SIS, e.g., pedestrians with various scales in the orange circles of Figure 7(f).

Moreover, Figure 8 and 9 shows the detection results by our unsupervised UMPD on both real-world [4, 8] and simulated datasets [12]. There are still some wrong results near the edges of region, where less overlapped camera views are insufficient for detection, especially on more populated MultiviewX in Figure 9(b). Such challenges remain to be tackled by unsupervised works in the future.

### 4.6 Comparisons with the State-of-the-arts

In Table 6, we compare our proposed unsupervised approach UMPD with fully-supervised state-of-the-art methods on Wildtrack, Terrace, and MultiviewX benchmarks: detection-based methods RCNN & Clustering [36] and POM-CNN [4]; anchor-based methods Deep-MCD [5] and Deep-Occlusion [2]; anchor-free methods MVDet [12], SHOT [31], MVDeTr [11], 3DROM [27], and MVAug [7]; cross-domain method GMVD [33]. "‡" are results by training on simulated

MultiviewX labels without real Wildtrack labels. For real-world datasets Wildtrack and Terrace, our proposed UMPD surpasses RCNN & Clustering, POM-CNN, Deep-Occlusion by MODA and MODP, as well as DeepMCD and GMVD by MODA, where UMPD better generalizes without any source domain. As is shown in Table 1 and Figure 9(b), MultiviewX has higher crowdedness. For this challenging dataset, UMPD out-performs RCNN & Clustering by MODA and MODP, as well as DeepMCD and Deep-Occlusion by MODP. In summary, our fully-unsupervised UMPD achieves competitive performances on all real-world and simulated datasets, without any manual labels essential to mainstream supervised methods.

### 5 Conclusion

In this paper, we have proposed a novel unsupervised multi-view pedestrian detection approach UMPD, which eliminates the heavy burden of manual annotations. For such a challenging task, three key components are proposed: SIS segments the PCA values of DINOv2 features iteratively, with zero-shot semantic capability of CLIP to identify pedestrians. Based on 2D-3D cross-modal mapping, GVD encodes multi-view images into a 3D volume, and learns to predict 3D density and color by the rendering losses with SIS pseudo labels. VBR constrains the predicted 3D density to be naturally vertical on the BEV ground plane. With these powerful components, our proposed UMPD achieves competitive performances on challenging benchmarks Wildtrack, Terrace, and MultiviewX. We hope this work, as the first fully-unsupervised method in this field, could be a start and inspire more interesting future researches.

## Acknowledgments

This work was supported by National Natural Science Foundation of China under Grants 62072032 and 62076024, and National Science Fund for Distinguished Young Scholars under Grant 62125601.

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
