# OpenReview forum: "Unsupervised Multi-view Pedestrian Detection"
_acmmm.org/ACMMM/2024/Conference — MM2024 Poster_

### Official Review · Reviewer_BNqS · 2024-05-24

**Rating:** 5
**Confidence:** 2

**Summary:**

This paper proposes a novel approach to pedestrian detection using multiple camera views without the need for manual annotations. The method, termed UMPD, leverages vision-language models and 2D-3D cross-modal mapping to detect pedestrians in 3D space. The experiments have demonstrated the acceptable performance of the method and the effectiveness of the three components.

**Strengths:**

1. The proposed UMPD is the first fully-unsupervised multi-view pedestrian detection method, addressing the challenge of manual annotation. Experimental results demonstrate its achievement of acceptable results compared to other non-fully-unsupervised methods.
2. The paper provides relevant code and results, making it more credible.
3. The article is well-written.

**Limitations:**

I'm not an expert in this field, but I believe the innovation and significance of this work are commendable. The only minor weakness is that the results in Table 7, while acceptable, are not particularly outstanding.

**Suitability:**

3

---

### Official Review · Reviewer_zAX2 · 2024-05-24

**Rating:** 2
**Confidence:** 3

**Summary:**

This paper presents UMPD, a method that eliminates the need for manual annotations in pedestrian detection using multiple camera views. UMPD consists of three components: Semantic-aware Iterative Segmentation (SIS) for generating 2D masks as pseudo labels, Geometry-aware Volume-based Detector (GVD) for predicting 3D density and color from multi-view images, and Vertical-aware BEV Regularization (VBR) to maintain vertical alignment in BEV projections. Evaluations on Wildtrack, Terrace, and MultiviewX datasets show UMPD performs competitively with state-of-the-art supervised methods. This approach is the first fully unsupervised method in the field, demonstrating significant potential for future research.

**Strengths:**

- The proposed approach represents a valuable adaptation of unsupervised learning to the multi-view pedestrian detection task, which is essential when considering the potential labeling costs from a real-world perspective.

- In conjunction with the CLIP model, the approach effectively leverages multi-modal knowledge to achieve notable performance.

**Limitations:**

- While this work (unsupervised/self-supervised) learning can be the first work in the multi-view pedestrian detection, the authors need to compare the existing unsupervised/self-supervised learning for the various multi-view tasks (e.g., [Ref_1, Ref_2, Ref_3]). If not, the effectiveness of the proposed unsupervised learning cannot be known.

- As mentioned in the first bullet in Sec. 1, one of the main novelty of this work is utilizing DINO and CLIP to generate pseudo-labels. However, this method is not novel. The previous works (e.g., [Ref_4] already adopt this scheme (difference between the existing works and this paper seems only the task-difference). What is the difference?

- The proposed method is heavily rely on the existing visual-encoder and text-encoder.

- The performance depends on the template used in the clip text encoder.

- In an unsupervised setting, the effectiveness of the proposed method relies on the choice of $\phi_{thr}$. The authors need to conduct experiments by varying $\phi_{thr}$.

- Why the authors does not adopt the recent unsupervised works for the multi-view pedestrian detection for comparison.

[Ref_1] X. Liu et al., "Unsupervised Multi-View Object Segmentation Using Radiance Field Propagation," in NeurIPS 2022.

[Ref_2] J. Li et al., "DS-MVSNet: Unsupervised Multi-view Stereo via Depth Synthesis," in ACM MM 2022.

[Ref_3] K. Xiong et al., "CL-MVSNet: Unsupervised Multi-view Stereo with Dual-level Contrastive Learning," in ICCV 2023.

[Ref_4] J. Kerr et al., "LERF: Language Embedded Radiance Fields," in ICCV 2023.

**Suitability:**

2

---

### Official Review · Reviewer_W6QU · 2024-05-27

**Rating:** 3
**Confidence:** 3

**Summary:**

This paper proposes a novel unsupervised multi-view pedestrian detection method, aiming to reduce the labor intensity of annotations. This method includes three main modules: 1) Semantic-aware Iterative Segmentation (SIS), 2) Geometry-aware volume-based Detector (GVD), and 3) Vertival-aware BEV Regularization (VBR). The authors conduct extensive experiments on the mainstream multi-view pedestrian detection datasets, such as Wildtrack, Terrace, and MultiviewX, demonstrating the competitive performance compared to previous fully supervised methods.

**Strengths:**

1.	Innovative: This paper proposes a completely unsupervised multi-view pedestrian detection method, which is an important innovation in the field of pedestrian detection.
2.	Depth of technology: UMPD combines multiple advanced technologies such as DINOv2, CLIP and 3D volume construction, showing deep technical research.
3.	Experimental validation: The authors conducted extensive experiments on multiple datasets to verify the effectiveness of the proposed method.
4.	Performance Competition: UMPD shows competitive performance compared to existing supervised learning methods, which indicates the potential of unsupervised learning in this area.

**Limitations:**

1.	The generalization ability of the model is not discussed in detail in the paper, especially in different environments and different pedestrian densities.
2.	For the factors that affect the accuracy of pedestrian detection, such as occlusion, illumination change and angle of view change, the paper lacks an in-depth analysis of the robustness of the model.
3.	How to determine the threshold used in subsection 3.1.1?
4.	The related metrics (e.g., MODA, MODP) should be clarified clearly.
5.	Figure 8 lacks the result visualization of the Terrace dataset.
6.	In formula (11), there is no corresponding weight coefficient for the loss function, and the ablation experiment for each weight of the loss function is lacking.
7.  The method is only evaluated on small datasets, such as Wildtrack and MultiviewX. I doubt whether this method can be performed on large datasets such as CVCS or CityStreet as in "Multi-View People Detection in Large Scenes via Supervised View-Wise Contribution Weighting".
8. There is no possible baseline methods for unsupervised settings.

**Suitability:**

2

---

### Meta-Review · Area_Chair_8wER · 2024-07-07

**Recommendation:** Accept (Poster)
**Confidence:** 5

**Metareview:**

Initially, the paper was met with interest due to its innovative approach to unsupervised pedestrian detection, a significant step forward in reducing the labor intensity of annotations. However, concerns were raised about the generalization ability of the model, the robustness against various environmental factors, and the lack of detailed analysis on these aspects. The initial average rating leaned towards acceptance with reservations.

Post-rebuttal, the authors addressed several concerns by providing additional clarifications on the method's dependency on existing technologies and its performance in unsupervised settings. They also committed to extending their experiments to larger datasets and including more recent methods for a fair comparison. Despite these improvements, some concerns about the method's robustness and generalization remained partially addressed. The final average rating shifted slightly to a weak accept due to the novelty and potential impact of the research.

AC noticed and considered the authors' email during the rebuttal. Looking forward to seeing the final improved version of this paper.

---

### Meta-Review · Senior_Area_Chairs · 2024-07-10

**Recommendation:** Accept (Poster)
**Confidence:** 5

**Metareview:**

This paper received mixed ratings initially. After rebuttal, all the reviewers tend to accept the paper. SAC and AC agree with reviewers and recommend acceptance of the paper.